# Nutrient Profiles of Commercially Produced Complementary Foods Available in Burkina Faso, Cameroon, Ghana, Nigeria and Senegal

**DOI:** 10.3390/nu15102279

**Published:** 2023-05-11

**Authors:** Asha Khosravi, Eleonora Bassetti, Katelyn Yuen-Esco, Ndeye Yaga Sy, Rosenette Kane, Lara Sweet, Elizabeth Zehner, Alissa M. Pries

**Affiliations:** 1Helen Keller International, New York, NY 10017, USA; askh85@hotmail.com (A.K.); eleonora.bassetti7@gmail.com (E.B.); kyuen@hki.org (K.Y.-E.); ezehner@hki.org (E.Z.); 2Helen Keller International, Dakar 12500, Senegal; nysy@hki.org (N.Y.S.); rkane@hki.org (R.K.); 3JB Consultancy, Johannesburg 2198, South Africa; lara@jbconsultancy.co.za

**Keywords:** complementary foods, infant and young child feeding, nutrient profiling, food labelling/standards

## Abstract

The nutritional quality of commercially produced complementary food (CPCF) varies widely, with CPCF in high-income settings often containing excessive levels of sugar and sodium. Little is known about the nutritional quality of CPCF available in the West Africa region, despite their potential to improve the nutrition of infants and young children (IYC). This study evaluated the nutritional quality of CPCF available in five West African countries using the WHO Europe nutrient profiling model (NPM) and assessed their suitability for IYC based on label information. The proportion that would necessitate a “high sugar” warning was also determined, and the micronutrient (iron, calcium, and zinc) content was assessed against IYC-recommended nutrient intakes. Of the 666 products assessed, only 15.9% were classified as nutritionally suitable for promotion for IYC. The presence of added sugar and excessive sodium levels were the most common reasons for a product to fail the nutrient profiling assessment. Dry/instant cereals contributed the highest percentage of recommended nutrient intake (RNI) per serving. This highlights the need for policies to improve the nutritional quality of CPCF in West Africa, including labeling standards and the use of front-of-pack warning signs to promote product reformulation and clearly communicate nutritional quality to caregivers.

## 1. Introduction

A child’s first 1000 days, from conception through the first two years of life, is a unique period when the foundations of optimum health, growth, and neurodevelopment across the lifespan are established. Infants have a limited gastric capacity and can only consume small quantities of food; therefore, it is crucial for complementary foods to be nutrient-dense, particularly in micronutrients essential for growth and development [1,2]. The World Health Organization (WHO) recommends the use of nutrient-rich, home-prepared, locally available foods for infant and young child feeding; however, in contexts where nutritionally adequate diets are limited, the use of nutritionally appropriate commercially produced complementary food (CPCF) could be one option for improving infant and young child (IYC) dietary adequacy and filling existing micronutrient gaps [3]. Research has highlighted the importance of fortified foods in addressing micronutrient needs, particularly in low- and middle-income country (LMIC) settings where diets based on traditional foods can be nutritionally lacking [4]. However, CPCFs vary widely in nutritional quality, necessitating the assessment of their adequacy for IYC feeding. Recent studies assessing the nutritional composition of CPCF available in high-income settings have shown excessive sugar and sodium content [5,6,7,8,9]. Given the establishment of lifelong taste preferences during infancy and young childhood [9,10], consumption of nutritionally inappropriate CPCF may predispose IYC to higher sugar and salt intakes in later childhood and into adulthood [11]. While the evidence is not yet conclusive, the consumption of CPCF high in sugar or salt may increase a child’s risk for overweight, obesity, and diet-related chronic disease [12]. Childhood consumption of free sugars through inappropriately formulated CPCF products also carries an increased risk of dental caries [13]. In addition, unhealthy foods can displace other more nutrient-dense foods and breastmilk, contributing to poor dietary adequacy [14,15].

The WHO *Guidance on Ending the Inappropriate Promotion of Foods for Infants and Young Children*, which was warmly welcomed as part of World Health Assembly Resolution 69.9, states in recommendation 3: “Foods for infants and young children that are not products that function as breast-milk substitutes should be promoted only if they meet all the relevant national, regional, and global standards for composition, safety, quality, and nutrient levels and are in line with national dietary guidelines” [16]. The WHO recommendation further encourages the use of nutrient profile models (NPM) to guide decisions on which foods for older infants and young children (6–23 months of age) are inappropriate for promotion. Nutrient profiling entails the classification of food items according to their nutritional composition to promote health and has been widely used as a tool to restrict or allow product promotion, particularly those marketed to children [17]. In 2019, the WHO Regional Office for Europe published an NPM for CPCF (WHO Europe NPM) to support the implementation of WHA 69.9 [15]. Globally, the WHO Europe NPM is the first, and currently the only, NPM developed specifically for CPCF products. The model includes nutritional composition requirements, as well as a front-of-pack warning for products with high total sugar content. The WHO Europe NPM has been validated for products in European markets [6,7,9] and applied to products available in the Southeast Asia region [18]. However, its application to other regions where CPCF is increasingly available is a necessary step in enabling the use of nutrient profiling for CPCF globally. 

Childhood malnutrition is widespread and persistent across West Africa [19], where undernutrition contributes substantially to child mortality. Stunting and wasting affect 31% and 7% of children under the age of five, respectively [20]. Concurrently, the region is also experiencing a nutrition transition where high levels of undernutrition are coupled with a growing burden of overweight/obesity and diet-related noncommunicable diseases (NCDs) [21], with the number of overweight children below five years of age increasing by nearly 24% since 2000 [22]. Exposure to high levels of sugar, fat, and sodium early in life can contribute to the growing prevalence of overnutrition in the region. 

Micronutrient deficiencies are a major public health concern across the West African region. The most prevalent deficiencies among infants and children under five include iron and zinc, with average prevalence rates of 72%, and 37%, respectively [20], as well as calcium deficiency [23]. These deficiencies can result in anemia, weakened immunity, and stunted growth, with serious health consequences [20]. Only 23% of older infants and young children across West Africa achieve minimum dietary diversity and 42% minimum meal frequency [20]. Given the significant prevalence of micronutrient deficiencies among children in West Africa [24,25,26] and the limited nutrient density of local complementary foods [27], consideration of the micronutrient content in CPCF is also critical for optimal older infant and young child nutrition. The consumption of CPCF is increasingly prevalent in West African countries, with 12% and 31% of infants 6–8 months in Cameroon and Ghana, respectively, consuming fortified “baby foods” and 49% of 6–23-month-olds in Dakar, Senegal, consuming any CPCF [28,29]. A study in urban Ghana estimated that children 6–12 months of age consumed an estimated 30 g of commercial ‘infant cereals’ approximately three times a day, while children 12–23 months consumed 45–60 g approximately two times a day [30]. Despite their increased use, there has been no assessment of the nutritional quality of CPCF available in the West African region.

This paper aims to assess the content of selected nutrients and nutrient profiles of CPCF available on the market in Burkina Faso, Cameroon, Ghana, Nigeria, and Senegal to better understand the nutritional appropriateness of CPCF for IYC in this region. The objectives are to: (1) determine the presence of added sugar and the content of total sugar, sodium, total fat, iron, calcium, and zinc declared on CPCF labels; (2) determine the proportion of CPCF that is nutritionally suitable/not suitable to be promoted for older IYC based on the WHO Europe NPM; (3) determine the proportion of CPCF that would require a “high sugar” warning based on the WHO Europe NPM; and (4) evaluate iron, calcium, and zinc content of CPCF against IYC recommended nutrient intakes.

## 2. Materials and Methods

### 2.1. Study Design

This study involved a cross-sectional analysis of CPCF label information on products identified in Burkina Faso, Cameroon, Ghana, Nigeria, and Senegal. CPCFs in this study were defined as products specifically marketed as suitable for feeding older infants and young children if they met at least one of the following criteria: (1) were recommended for introduction at an age less than three years; (2) were labelled with the words “baby”, “infant”, “toddler”, “young child” or a synonym; (3) had a label with an image of a child who appeared to be younger than three years of age or who was feeding with a bottle; or (4) were in any other way presented as being suitable for children under three years of age [31].

### 2.2. Product Identification and Data Management

For Burkina Faso, Cameroon, Ghana, and Nigeria, databases of CPCF identified at points of sale were purchased from the market research company, Innova Market Insights (IMI). These databases included CPCF identified as newly launched products by IMI research staff, who conduct weekly visits to purposively sampled retailers in cities across each country. Retailers in these four countries included supermarkets, neighborhood boutiques, pharmacies, and roadside kiosks. The databases were sourced from IMI in February 2022, and because only newly launched products are added to these databases, the entirety of years of IMI data collection was included in this study. This, therefore, included products identified since February 2021 in Burkina Faso, since December 2013 in Cameroon, since February 2012 in Ghana, and since February 2012 in Nigeria. These datasets provided a cumulative account of all products that had been on the market since these points in time. After products were identified, label information was extracted by IMI and entered into the databases, with information translated into English when necessary. Photographs of products were also taken and uploaded to the databases. Entered product label information was checked by local and regional IMI editors. After purchasing from IMI, each product’s label information was also cross-checked against the product photographs by a study author to ensure label information accuracy.

For Senegal, store scoping was conducted to obtain a wide variety of all CPCF products available for purchase in Dakar and Guédiawaye Departments during May–June 2021. In Dakar Department, local researchers compiled a list of larger stores, including supermarkets, hypermarkets, and pharmacies. The independent stores identified were exhaustively sampled. Among chain retailers, the store that stocked the greatest variety of CPCF products was purposively sampled. In the peri-urban Guédiawaye Department, store scoping found that CPCF points of sale included only one larger store (a supermarket), and four types of smaller stores (superettes, small pharmacies, gas station boutiques, and neighbourhood boutiques). Only a small number of superettes and gas station boutiques were identified across all communes and were thus exhaustively sampled. Each commune was found to have numerous pharmacies and neighbourhood boutiques, and so the two largest pharmacies and neighbourhood boutiques per commune were purposively sampled. A total of 10 larger stores were visited in the Dakar Department, and 31 stores (1 larger, 30 smaller) were visited across the five communes of the Guédiawaye Department. CPCF products were purchased in sampled stores so label information could be extracted for analysis. One of each unique CPCF product was purchased from the first store at which it was encountered. Products carrying the same brand name but different sub-brands, descriptive names, age recommendations, and age categories, or made by different manufacturers, were treated as unique products and purchased. Different flavours of the same product were also treated as unique products as their nutrient content could vary. CPCFs that only provided nutrition information or an ingredient list in a language other than English, French, or Wolof (the local language of Senegal) were excluded from the study. Label information provided in French/Wolof took precedence over the English text. General product information, nutrient content declarations, and ingredient lists were entered into a Microsoft Excel database. Local researchers translated the text from French/Wolof to English when necessary.

For all five countries, the following information was extracted from labels: brand name, manufacturer, ingredient lists, nutrient content per 100 g/serving (energy, protein, total fat, total sugar, sodium, iron, calcium, zinc), and serving size. Nutrient contents were calculated per 100 g where contents were declared as per serving. The presence of added sugars/sweeteners was identified within ingredient lists, as well as the percentage of fruit, protein, and water content for relevant product categories. The country and region of the manufacturer were also identified based on label information or an online search for the manufacturer.

### 2.3. Analysis

The WHO Europe NPM contains two components to assess if a CPCF product is suitable for promotion:assessment of nutrient composition and assessment of labelling practices. As the aim of this study was to assess the nutritional suitability of CPCF, only the nutrient composition component was used. Product names and ingredients were first reviewed, and all CPCFs were placed in one of five overall product categories (16 subcategories) outlined in the WHO Europe NPM (Table 1). Products that did not fit into these categories were excluded from the study. According to the WHO Europe NPM, product categories 4.1 (confectionery, sweet spreads, and fruit chews) and 5. (juices and other drinks) should not be marketed as suitable for infants and young children, regardless of their nutritional content. These categories were therefore not assessed against the nutrient composition requirements and were classified as having automatically failed the NPM.

After product categorization, the extracted CPCF label information, including ingredient lists and nutrient content declarations, was assessed against category-specific nutrient composition requirements. In cases where product labels were missing nutrient content information, these products were unable to pass that specific nutrient assessment. A product was classified as nutritionally suitable for older infants and young children if it met all category-specific nutrient requirements. Separate from the nutrient profiling evaluation, the WHO Europe NPM also evaluates whether a product should carry a front-of-pack “high sugar” warning based on total sugar content. Nutrient profiling assessment based on the WHO Europe NPM was conducted using pre-designed Microsoft Excel spreadsheets developed by a team of researchers at Leeds University’s Nutritional Epidemiology Group in collaboration with the WHO Regional Office for Europe.

The categorization of products as fortified was based on a review of ingredient lists for the addition of minerals/vitamins. Assessment of iron, zinc, and calcium content of CPCF was based on methods used by Dreyfuss et al., whereby the contribution of micronutrient content to age-specific recommended nutrient intakes (RNI) by serving size was calculated [32]. For most products, calcium/iron/zinc content per 100 g was provided on the label. For products that provided nutrient content as a percentage of recommended dietary allowance (RDA) without also providing the nutrient content by weight, calcium/iron/zinc content were calculated by multiplying the reported percentage of the RDA contained in one recommended serving (as listed on the product label) by the RDA for the product’s recommended age of use. RDAs from the United States were used for this calculation, as the products that reported micronutrient content as a percentage of RDA were manufactured in the United States [33]. If no recommended age was provided on the label, the micronutrient content could not be calculated, and the product was excluded from this component of the analysis. If the product’s recommended age of use spanned more than one age category for the specified country’s RDA, then the average of the RDA values from the two age categories was used [32]. Declared nutritional information for the three micronutrients was input into Microsoft Excel to assess their levels per serving size. To calculate the median percentage contribution to RNI by serving size for each category of products, the calcium/iron/zinc content per serving size was divided by the age-specific RNI. A value greater than 100% indicated that a product’s micronutrient content was greater than the reference RNI for a specific age range. A percentage less than 100% indicated that a product’s micronutrient content was lower than the reference RNI for a specific age range. The WHO/FAO nutrient recommendations were used for this standardized comparison, with an assumed 10% bioavailability for iron and moderate bioavailability for zinc [34]. The RNI values used for this analysis can be found in Appendix A.

Statistical analysis was conducted using Stata 14. Descriptive statistics were calculated and summarized using proportion and medians with interquartile ranges (IQR) for nonnormally distributed data. Differences in proportions of products were tested using the Pearson chi-square test, with significance defined as *p* < 0.05.

## 3. Results

Across the five countries included in this study, a total of 714 CPCF products were identified; 48 products were excluded, resulting in a final count of 666 products (Figure 1). The greatest number of products included were from Senegal (*n* = 352), Ghana (*n* = 131), and Nigeria (*n* = 111), with these countries also having the largest number of product categories: 13 in Senegal, 10 in Ghana, and 8 in Nigeria. Of the 666 CPCF products in the study, 641 were assessed against the WHO Europe NPM for nutritional suitability. For Burkina Faso and Cameroon, all products were assessed against the WHO Europe NPM, while 25 products from Ghana, Nigeria, and Senegal were automatically classified according to the WHO Europe NPM as not nutritionally suitable due to their product category type (i.e., juices and other drinks) (Figure 1).

Dry/instant cereals and soft-wet spoonable, ready-to-eat foods (purees) were the most prevalent overall product categories, making up 41.3% (*n* = 275) and 43.4% (*n* = 289), respectively, of all CPCF products included in the study. Within purees, fruit purees were the most predominant, accounting for nearly half (47.1%, *n* = 136) of all purees. Chunky meals were not common across the five countries, making up only 7.4% (*n* = 49) of all CPCF products, and were only found in Senegal. Dry snacks, finger foods, and juices/other drinks were the least common, making up 4.2% (*n* = 28) and 3.8% (*n* = 25) of all products, and these products were only found in Ghana, Nigeria, and Senegal. No sweet confectionary/sweet spreads/fruit chews or fruit snacks were identified across any of the five countries.

Of the 641 products assessed by the WHO Europe NPM, only 15.9% (*n* = 102) met all relevant nutrient composition requirements (Table 2), including 22.2% (*n* = 64) of purees, 14.3% (*n* = 7) of chunky meals, and 11.3% (*n* = 31) of dry/instant cereals. Snacks/finger foods performed the most poorly, with none of these products meeting all relevant nutrient composition requirements.

Of the 641 products assessed by the WHO Europe NPM, over half (56.6%, *n* = 363) were manufactured in Europe, approximately one-third (32.0%, *n* = 205) in Africa, and the remaining in South America (5.8%, *n* = 37), North America (3.4%, *n* = 22), Asia (1.2%, *n* = 8) and the Middle East (0.9%, *n* = 6). Products manufactured in Africa, Europe, and Asia performed similarly in the WHO Europe NPM, with only 16.6% (*n* = 34), 15.7% (*n* = 57), and 12.5% (*n* = 1), respectively, meeting all nutrient composition requirements, while 44.5% (*n* = 10) of products manufactured in North America achieved this. No products manufactured in South America or the Middle East passed the WHO Europe NPM. Results of the WHO Europe NPM assessment by country are detailed in Appendix A.

While the majority of dry/instant cereals met the fat, protein, fruit content, and sodium requirements, less than one-quarter (24.0%, *n* = 66) met the requirement for no added sugar/sweetener. Almost half of all dry/instant cereals that declared total sugar content would warrant a “high sugar” warning on their labels (Figure 2), and the median total sugar content of dry/instant cereals across the five countries ranged from 25.1–32.2 g per 100 g product (Appendix A). While only one-fifth (22.2%, *n* = 64) of purees met all relevant nutrient composition requirements, performance varied depending on the sub-category type of puree assessed. Whereas the majority (72.0%, *n* = 208) of purees met the no added sugar/sweetener requirement, the exception was dairy-based desserts and cereal purees, of which only 8.3% (*n* = 4) contained no added sugar. Despite most purees containing no added sugar, the total sugar content was problematic for some sub-category types, with 93.3% (*n* = 112) of fruit purees and almost half (43.5%, *n* = 10) of vegetable and cereals purees warranting a “high sugar” FOP warning (Figure 2).

Only one-third (*n* = 2) of pureed meals with meat/fish and one-quarter (*n* = 10) of pureed meals with cheese met the sodium requirement. Most purees across all sub-category types met the fruit content and total fat requirements. All chunky meals met the no added sugar/sweetener, fruit content, and total fat requirements, but only one-fifth (20.4%, *n* = 10) and just over one-third (34.7%, *n* = 17) met the sodium and protein requirements, respectively.

The median sodium content of chunky meals ranged from 97–99 mg per 100 g product (Appendix A). Sugar content drove the poor performance of all snacks/finger foods; only 3.6% (*n* = 10) of snacks/finger foods did not contain added sugar/sweeteners and only 10.7% (*n* = 3) of snacks/fingers foods that declared total sugar content contained less than 15% of energy from sugar content. The median total sugar content of snacks/finger foods across the three countries where these products were found ranged from 17.3–28.0 g per 100 g product (Appendix A).

Among all 666 products included in the study, 40.2% (*n* = 268) were fortified. The fortification status of products varied substantially by country, from 22.4% (*n* = 79) of CPCF in Senegal to 90.0% (*n* = 18) of CPCF in Burkina Faso. This was primarily driven by the distribution of product categories available in each country, as dry/instant cereals were the most commonly fortified (85.1%, *n* = 234), followed by snacks/finger foods (64.2%, *n* = 18) while only 5.2% (*n* = 15) of purees and 2.0% (*n* = 1) of chunky meals were fortified. Among dry/instant cereals, a greater proportion of fortified products (*n* = 234) met all relevant nutrient composition requirements as compared to non-fortified (*n* = 41) products (12.4% versus 4.9%, respectively, *p* < 0.001), while more non-fortified purees (*n* = 274) met all nutrient composition requirements as compared to fortified (*n* = 15) purees (23.4% versus 0.0%, respectively, *p* = 0.063). Nearly half of fortified dry/instant cereals (46.7%, *n* = 56) and one-third of fortified purees (33.3%, *n* = 5) had total sugar content that warranted a “high sugar” warning. The majority of fortified and non-fortified snacks/finger foods (87.5% [*n* = 14] and 87.5% [*n* = 7], respectively) warranted a “high sugar” warning.

Across all five countries, the majority of dry/instant cereals reported micronutrient content on their labels; two-thirds (66.1%, *n* = 179) reported calcium content, 77.9% (*n* = 211) reported iron content, and 53.9% (*n* = 146) reported zinc content. Despite being prevalent product categories in some countries, few purees, chunky meals, or snacks/finger foods declared micronutrient content. For example, 194 purees were identified in Senegal, but only 10.3% (*n* = 20) declared calcium content, 0.0% (*n* = 0) iron content, and 1.5% (*n* = 3) zinc content. The median calcium, iron, and zinc content of CPCF by product category across each of the five West African countries are detailed in Appendix A. Median content per 100 g of dry/instant cereals was relatively similar for the three micronutrients across all five countries, but variable for purees and snacks/finger foods. The variation among purees was likely driven by different sub-categories of purees in each country whose ingredients would vary in micronutrient content. Dry/instant cereals and snacks/finger foods contained the highest quantities of calcium, iron, and zinc per 100 g of product.

The median percentage of WHO/FAO RNI for calcium, iron, and zinc per serving met by CPCF identified across the five countries is presented in Table 3. Dry/instant cereals met the highest percentage of RNI per serving for calcium, iron, and zinc. The highest calcium concentrations were found in dry/instant cereals from Cameroon and Burkina Faso, which provided more than half (56% and 57%, respectively) of the RNI for calcium for older infants and slightly under half (45% and 46%, respectively) for children aged 12 to 36 months. Across all countries, a median serving of dry/instant cereals provided roughly two-thirds of the RNI for iron for children aged 12–36 months and 35–41% for older infants, and 34–58% of the zinc RNI for children aged 6–36 months. Despite a median serving size of 100 g or more, purees met only 14–33% of the RNI for calcium for older infants and 11–26% for children aged 12–36 months. Iron and zinc content was rarely declared on puree labels. With relatively small portion sizes, snacks/finger foods provided only 6–12% and 5–9% of the RNI for calcium for older infants and children aged 12–36 months, respectively, and provided less than a quarter (7–24%) of recommended iron to young children and even less (4–15%) to older infants aged 6–12 months.

## 4. Discussion

This study assessed the nutritional suitability of CPCF available on the market in Burkina Faso, Cameroon, Ghana, Nigeria, and Senegal using the WHO Europe NPM. Of 666 products included from the five countries, 25 were automatically classified as not suitable based on their product category, and of the 641 products assessed by the NPM, only 102 met all relevant nutrient composition requirements; overall, 15.3% of all products were classified as nutritionally suitable for promotion for IYC. To our knowledge, this is the first study to nutrient profile CPCF in the West Africa region and the first application of the WHO Europe NPM within the African continent. Evaluation of iron, calcium, and zinc content showed that some CPCF, particularly dry/instant cereals, would potentially provide substantial contributions of these micronutrients to RNIs for IYC. These results indicate that while some product categories of CPCF may serve as a source of essential micronutrients, CPCF in West Africa also contains concerning levels of nutrients inappropriate for IYC diets.

The presence of added sugar and high sodium content were the most common reasons for CPCF being flagged as not nutritionally suitable for IYC. Only 50.6% of all products met the WHO Europe NPM requirement for no added sugar/sweeteners. Further, nearly half of all products that declared total sugar content warranted a “high sugar” warning. These findings are consistent with a recent study in Europe [6], which found almost one-third of CPCF across ten European countries contained added sugar and 30% of total energy came from sugar, far exceeding the WHO recommended limit of 10% for older children [35]. Similarly, a study in South Africa reported that 80% of CPCF cereals and pureed desserts contained added sugar, and over three-quarters of all CPCFs had high sugar levels [8]. In our West African study, fruit purees performed particularly poorly on sugar content, with over 90% warranting a “high sugar” warning. While the sugar content of fruit puree is primarily intrinsic and not from added sugars, the intense pureeing process used to make these CPCF products release intrinsic sugars from the cell walls of fruit and vegetables, resulting in readily available and problematic free sugars [35].

As a high level of free sugar intake is associated with an increased risk of being overweight and developing NCDs [36], the presence of such high levels of free sugars in purees is a significant concern for the African continent, where rates of overweight/obesity and diet-related NCDs are growing, including hypertension [37] and diabetes [38]. Research has also shown that added sugar intake in infancy is associated with higher added sugar intake in later childhood due to the establishment of taste preferences during this period of early life [11], and evidence suggests that high sugary food consumption among children raises their risk of high blood triglycerides and is linked to a higher incidence of dental caries [39]. While the majority of CPCF met the sodium requirements, over half of all savoury meals/purees and snacks/finger foods exceeded the maximum sodium requirement. Similar findings have been found in Europe [15] and the United States [40], where most savoury meals and snack products currently on the market would require reformulation to reduce sodium levels. A high salt intake throughout childhood is associated with high blood pressure [41] and the establishment of a preference for salty foods [42]. According to global guidance and national dietary guidelines, sugars, fruit juice, concentrated fruit juice, and other sweeteners should not be used in IYC diets, and salt intake should be limited [15,43]. The presence of added sugars and high levels of sodium in commercial products marketed for IYC in West Africa is cause for concern and highlights the need for product reformulation by manufacturers and strengthened product standards by national governments [7]. Infant feeding interventions, such as the promotion of exclusive breastfeeding for the first six months of life, followed by nutrient-rich complementary feeding, have the potential to reduce the prevalence of rising obesity and chronic diseases in West Africa and improve the health outcomes of infants and young children in the region [44].

Snacks/finger foods were the worst performing CPCF category assessed in this study, with no products meeting all WHO Europe NPM requirements, only 3.6% containing no added sugar/sweeteners, and just one-third meeting the sodium requirement. While snacks/finger foods were not commonly available across all five West African countries in this study, they did make up 13% of products in Nigeria and are becoming increasingly predominant in CPCF markets in many regions [18]. The growth of this category of CPCF has been noted in high-income contexts; an assessment of CPCF products available in the United Kingdom found that the proportion of CPCF which were snack foods increased from 10% in 2013 to 21% in 2019 [45]. This growth in the CPCF snack product market likely stems from the normalization of commercial snack food product consumption among children globally, as studies suggest that commercial snack food products are becoming an ever more prevalent part of IYC diets in LMICs [14]. A 2019 study estimated that 39%, 42%, 48%, and 62% of 6–23-month-olds in urban Burkina Faso, Cote d’Ivoire, Mali, and Niger had consumed a commercial snack food product in the previous 24 hours [46]. In West Africa, while only 2.7% of children under five years of age are overweight [20], the African continent is facing a growing problem of overnutrition, with 1 in 4 children overweight or obese [47]. Africa is facing a growing double burden of malnutrition, experiencing a slow decline in stunting and a fast rise in obesity due to economic development, urbanisation, increased international trade, and changing dietary habits [48]. The increase in ultra-processed food availability and consumption is linked with NCDs and increased risk of overweight and obesity as identified by cohort studies across the world [49].

Dry/instant cereals were the most commonly fortified CPCF across the five West African countries, with recommended serving sizes of dry/instant cereals contributing the largest proportion to RNI for the three micronutrients considered in this study, ranging from one-third to two-thirds of RNI for iron and zinc. Prior research from the region has also evaluated the compliance of these CPCF products with global fortification recommendations. A study assessing the nutrient composition, based on laboratory analysis and adequacy of 32 imported and locally produced cereal-based CPCF products in West Africa, found iron content of 75% of products conformed with the Codex Alimentarius specifications; however, calcium and zinc contents were far below the recommended levels [50]. The use of fortified CPCF for complementary feeding has been recommended as a means to improve dietary adequacy among IYC in LMIC [51], where nutrient density in the general diet can be limited [52]. A study examining micronutrient intake from cereal-based CPCF in Ghana found that the estimated daily consumption of such products contributed approximately 70% of the RNI for iron, zinc, and calcium [30]. In peri-urban South Africa, consumption of fortified cereals among 6-month-olds provided over 90% of total iron intake and over 50% of total zinc intake [53]. While such studies indicate an important potential role for CPCF in improving micronutrient intakes among IYC, it should be noted that there is limited evidence on whether CPCF reduces the risk of micronutrient deficiencies [54]. Additionally, several studies have shown that some CPCFs, particularly imported products, are unaffordable for poorer populations in West Africa, and there may be a risk that cereal-based CPCFs will be over-diluted or used irregularly [50]. Finally, the ability of fortified cereal-based CPCF to improve nutrient adequacy among IYC depends on the accurate fortification and labelling of these products. A recent study assessing the accuracy of label-declared nutrient contents of CPCF commonly available in Cambodia, Indonesia, and Senegal revealed a substantial discrepancy in micronutrient content; just over half of products had accurate label values for calcium, iron, and zinc compared to laboratory analysis [55]. Similarly, a study of 108 cereal-based CPCFs from 22 LMICs found discrepancies between label-declared values and laboratory-assessed values for protein and fat content [56]. While CPCF may have a role in improving dietary intakes among IYC in LMIC, these products must be appropriately formulated, affordable, and accurately labelled to ensure equitable access and informed choices for caregivers. Additionally, given that almost half of fortified dry/instant cereals included in this study warranted a “high sugar” warning, reformulation to reduce the total sugar and sodium content of many such products is also needed.

Global guidance proposes double-duty actions, aimed to address all forms of malnutrition (undernutrition, micronutrient deficiencies, overweight), and diet-related non-communicable diseases, as a means to tackle malnutrition more effectively [44]. In the context of many LMICs undergoing rapid nutrition transitions and the abundance of foods high in fat, sugar, and sodium during the first 1000 days of life, adequate early-life nutrition and appropriate complementary feeding can reduce the risk of undernutrition, obesity, and diet-related NCDs. Policies focused on improving the nutritional quality and labelling of CPCF can target all forms of malnutrition by directing consumers away from unhealthy products and towards healthier products for IYC [44]. Nutrient profiling can be a useful tool for such policies, aiding policymakers in determining which foods should not be promoted for IYC and encouraging product reformulation among manufacturers. In West Africa, where undernutrition is chronic and the nutrient density of homemade complementary foods can be limited, it is critical for CPCF products to be nutrient-dense. A nutrient profiling model for this region would therefore benefit from the inclusion of micronutrient requirements, particularly for those micronutrients where there are often nutrient gaps in diets. In light of the poor nutritional quality of most CPCFs assessed in this study, parents and caregivers in West Africa will require guidance in understanding the potentially harmful effects of inappropriate CPCFs [14,57,58]. The use of nutrient profiling models for placement of front-of-pack warning signs on products with excessive sugar, sodium, and unhealthy fat content is one tool by which poor nutritional quality of such products would be clearly communicated to IYC caregivers [59].

To our knowledge, this was the first study to apply the WHO Europe NPM to CPCF from an African region. In lieu of an NPM for CPCF specific to West Africa, this model was used as a first step to gauge its appropriateness and enable nutrient profiling of these products in the region. Overall, nutrient profiling products from West Africa with the WHO Europe NPM was successful, however, several limitations were noted. First, several local products did not fit into the predefined categories of the WHO Europe NPM and had to be excluded from the study. These included condiments (i.e., fish powders) and bottled water. As CPCF vary widely in nutrient content and composition across countries and product types, the exclusion of local products may underestimate the nutritional adequacy of local CPCF and hinder their comparison and classification.

In addition, several locally produced dry products contained plant-based proteins (i.e., peanut, soy, cowpeas); however, the WHO Europe NPM does not have any category for plant/legume-based products, making product categorization challenging. These products were ultimately categorized as dry/instant cereals as they also contained cereals/starches. A recent study implementing the WHO Europe NPM in Southeast Asia similarly noted challenges with product categorization and adaptations that were made to deal with additional product categories [18]. Any future NPM for the West African region should be adapted for this context and the range of products available. Additionally, products presented varying levels of label completeness for nutrient declarations, serving sizes, and preparation instructions. Missing label information precluded some products from having certain nutrient requirements assessed and prevented their inclusion in the micronutrient content assessment. CPCF marketing and labeling are not well regulated or monitored, leading to potential misinformation on nutrient content, ingredients, and serving size. This highlights the need for national governments to ensure comprehensive nutrient declarations on labels; incomplete nutrient content information means caregivers are not fully informed of the nutritional quality of products for IYC. Finally, given the high level of micronutrient deficiency in West Africa, an adapted version of the WHO Europe NPM would benefit from the inclusion of nutrient requirements for critical micronutrients such as iron, zinc, and calcium.

## 5. Conclusions

This study found that the majority of CPCF available in Burkina Faso, Cameroon, Ghana, Nigeria, and Senegal are not nutritionally suitable for promotion for IYC, with products often containing added sugars and high levels of sugar and sodium. These significant findings support the WHO’s request for governments to take steps to halt the inappropriate promotion of foods for older infants and young children in the interest of public health. Nutrient profiling is a tool that can be used to facilitate these actions, as well as spur product reformulation and the strengthening of national compositional standards for CPCF. An adapted version of the WHO Europe NPM can serve as a benchmarking tool to assess the quality of nutritional composition and labelling practices of CPCF in the West Africa region. Nutrient profiling can assist manufacturers in targeting nutrients of concern for product reformulation, as well as assist governments in determining which products are not suitable for marketing to IYC and whose nutritional quality may need to be communicated to caregivers through front-of-pack warning labels. Monitoring the nutritional quality of the food supply, in addition to complimentary measures such as adequate labelling information, marketing restrictions, and manufacturer accountability and guidance for product reformulation, can contribute to a supply of healthier CPCF.

## Figures and Tables

**Figure 1 nutrients-15-02279-f001:**
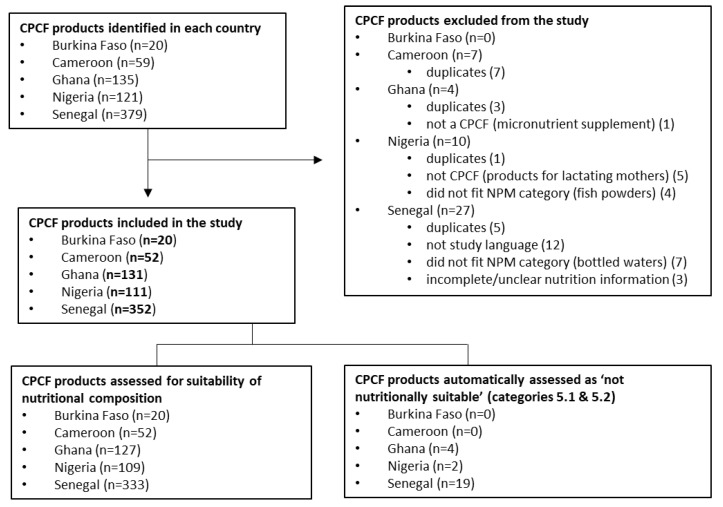
Flowchart of commercially produced complementary food (CPCF) product inclusion, exclusion, and assessment.

**Figure 2 nutrients-15-02279-f002:**
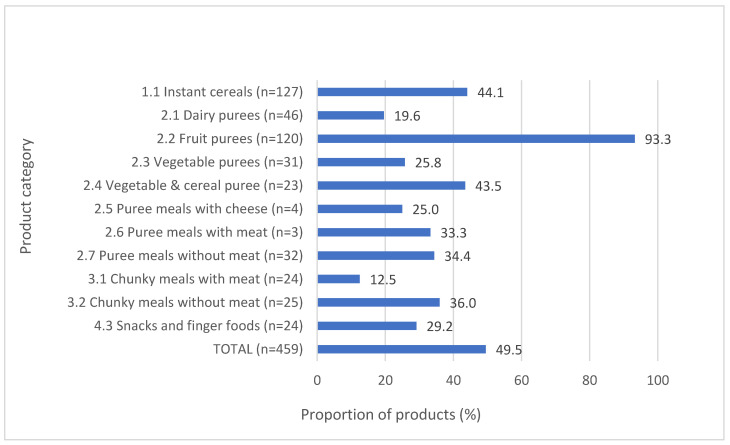
Proportion of products requiring a “high sugar” front-of-pack warning label.

**Table 1 nutrients-15-02279-t001:** WHO Europe NPM product categories.

Category 1: Dry, powdered, and instant cereal/starchy food
Category 1.1 Dry or instant cereals/starches
Category 2: Soft–wet spoonable, ready-to-eat foods
Category 2.1 Dairy-based desserts and cereal products
Category 2.2 Fruit purée with or without the addition of vegetables, cereals, or milk
Category 2.3 Vegetable only purée
Category 2.4 Puréed vegetables and cereals
Category 2.5 Puréed meal with cheese (but not meat or fish) mentioned in the name
Category 2.6 Puréed meal with meat or fish mentioned as the first food in the product name
Category 2.7 Puréed meals with meat or fish (but not named first in the product name)
Category 2.8 Purées with only meat, fish, or cheese in the name
Category 3: Meals with chunky pieces
Category 3.1 Meat, fish, or cheese-based meal with chunky pieces
Category 3.2 Vegetable-based meal with chunky pieces
Category 4: Dry finger foods and snacks
Category 4.1 Confectionery, sweet spreads, and fruit chews
Category 4.2 Fruit (fresh or dry whole fruit or pieces)
Category 4.3 Other snacks and finger foods
Category 5: Juices and other drinks
Category 5.1 Single or mixed fruit juices, vegetable juices, or other non-formula drinks
Category 5.2 Cow’s milk and milk alternatives with added sugar or sweetening agent

**Table 2 nutrients-15-02279-t002:** WHO Europe NPM nutrient composition assessment of commercially produced complementary food products available in Burkina Faso, Cameroon, Ghana, Nigeria, and Senegal ^1^.

Product Category	*n*	Met All Relevant Nutrient Requirements	No Added Sugar/Sweetener ^2^	Low/No Added Fruit ^3^	>15% Energy from Sugar ^4^	Met Sodium Requirement ^5^	Met Energy Density Requirement ^6^	Met Protein Requirement ^7^	Met Total Fat Requirement ^8^
**Dry, powdered, and instant cereal/starchy foods**									
1.1 Dry or instant cereals/starches	275	11.3 (31)	24.0 (66)	81.5 (224)	NA	73.8 (203)	NA	89.1 (245)	89.1 (245)
**Soft–wet spoonable, ready-to-eat foods (purees)**									
2.1 Dairy-based desserts and cereal products	48	0.0 (0)	8.3 (4)	79.2 (38)	NA	70.8 (34)	91.7 (44)	93.8 (45)	100.0 (48)
2.2 Fruit puree	136	23.5 (32)	73.5 (100)	NA	NA	95.6 (130)	36.8 (50)	NA	97.8 (133)
2.3 Vegetable-only puree	36	22.2 (8)	100.0 (36)	100.0 (36)	NA	66.7 (24)	NA	NA	94.4 (34)
2.4 Vegetable puree with cereals	25	32.0 (8)	100.0 (25)	92.0 (23)	NA	64.0 (16)	56.0 (14)	NA	100.0 (25)
2.5 Pureed meal with cheese	4	50.0 (2)	100.0 (4)	100.0 (4)	NA	25.0 (10)	75.0 (3)	75.0 (3)	100.0 (4)
2.6 Pureed meal with meat/fish in product name	6	16.7 (1)	100.0 (6)	100.0 (6)	NA	33.3 (2)	50.0 (3)	83.3 (5)	100.0 (6)
2.7 Pureed meal with meat/fish not in product name	34	38.2 (13)	97.1 (33)	94.1 (32)	NA	64.7 (22)	50.0 (17)	97.1 (33)	100.0 (34)
**Meals with chunky pieces**									
3.1 Chunky meal with meat/fish/cheese	24	4.2 (1)	100.0 (24)	100.0 (24)	NA	16.7 (4)	NA	8.3 (2)	100.0 (24)
3.2 Chunky meal with vegetables	25	24.0 (6)	100.0 (25)	100.0 (25)	NA	24.0 (6)	NA	60.0 (15)	100.0 (25)
**Dry finger foods and snacks**									
4.3 Snacks and finger foods	28	0.0 (0)	3.6 (10)	NA	10.7 (3)	39.3 (11)	NA	NA	60.7 (17)
**All products**	641	15.9 (102)	50.6 (324)	86.4 (412)	10.7 (3)	70.7 (453)	51.8 (131)	83.7 (348)	92.8 (595)

^1^ Values are presented as % (*n*); NA = not applicable based on category. ^2^ The following were considered added sugar/sweetener: sugar, juice (except lemon/lime), sucrose, dextrose, fructose, glucose, maltose, galactose, trehalose, syrup, nectar, honey, malted barley, malt extract, molasses. ^3^ Requirement definition per applicable category—1.1: <10% by weight dried/powdered fruit; 2.1/2.5/2.6/2.7/2.8/3.1/3.2: ≤5% by weight fruit puree; 2.3/2.4: no added fruit/fruit purée. Not applicable to categories 2.2 and 4.3; total products assessed = 477. ^4^ Applicable to category 4.3 only and among products declaring total sugar content. Total products assessed = 28. ^5^ Requirement definition per applicable category—1.1: sodium < 50 mg/100 kcal; 2.1/2.2/2.3/2.4/4.3: sodium < 50 mg/100 kcal and <50 mg/100 g; 2.5: sodium < 100 mg/100 kcal and 100 mg/100 g; 2.6/2.7/2.8/3.1/3.2: sodium < 50 mg/100 kcal and <50 mg/100 g (or <100 mg/100 kcal and <100 mg/100 g if cheese is listed in front- of-pack name). ^6^ Requirement definition per applicable category—2.1/2.2/2.4/2.5/2.6/2.7: energy density ≥ 60 kcal/100 g; total products assessed = 253. ^7^ Requirement definition per applicable category—1.1: <5.5 g/100 kcal total protein; 2.1/2.5/: ≥2.2 g dairy protein/100 kcal; 2.6: total protein ≥ 4 g/100 kcal from the named source and protein named as the first food(s) in the product name must be ≥10% by weight of the total product; 2.7: total protein ≥ 3 g/100 kcal and protein source mentioned in the product name must be ≥8% by weight of the total product; 2.8: ≥7 g/100 kcal total protein; 3.1: total protein ≥ 4 g/100 kcal and protein source mentioned in the product name must be ≥10% by weight of the total product; 3.2: ≥3 g/100 kcal total protein; total products assessed = 477. ^8^ Requirement definition per applicable category—1.1/2.1/2.2/2.3/2.4/2.7/3.2/4.3: ≤4.5 g/100 kcals total fat; 2.5/2.6/2.8/3.1: ≤6 g/100 kcal total fat.

**Table 3 nutrients-15-02279-t003:** Median percentage of WHO/FAO RNI for calcium, iron, and zinc met by a serving of CPCF ^1^.

Product Category	Calcium Content Declared (*n*)	Calcium RNI for 6–12 m	Calcium RNI for 12–36 m	Iron Content Declared (n)	Iron RNI for 6–12 m	Iron RNI for 12–36 m	Zinc Content Declared (n)	Zinc RNI for 6–36 m
**Burkina Faso**								
1. Dry, powdered, and instant cereal/starchy food (*n* = 20)	18	57% (40–75%)	46% (32–60%)	18	41% (29–54%)	66% (47–86%)	14	52% (40–61%)
**Cameroon**								
1. Dry, powdered, and instant cereal/starchy food (*n* = 46)	35	56% (42–74%)	45% (34–60%)	38	41% (27–54%)	65% (43–87%)	26	58% (47–67%)
2. Soft-wet spoonable, ready-to-eat foods (purees) (*n* = 6)	2	33% (32–34%)	26% (25–27%)	1	20% (--)	32% (--)	0	-
**Ghana**								
1. Dry, powdered, and instant cereal/starchy food (*n* = 54)	26	33% (23–56%)	27% (18–45%)	35	41% (34–52%)	66% (54–84%)	28	44% (33–63%)
2. Soft-wet spoonable, ready-to-eat foods (purees) (*n* = 69)	8	20% (4–27%)	16% (3–21%)	5	19% (3–40%)	30% (4–65%)	2	44% (42–46%)
4. Dry finger foods and snacks (*n* = 4)	1	6% (--)	5% (--)	1	4% (--)	7% (--)	0	-
5. Juices and other drinks (*n* = 4)	0	-	-	0	-	-	0	-
**Nigeria**								
1. Dry, powdered, and instant cereal/starchy food (*n* = 75) ^2^	49	46% (24–63%)	36% (19–50%)	56	35% (32–55%)	56% (52–88%)	37	34% (27–37%)
2. Soft-wet spoonable, ready-to-eat foods (purees) (*n* = 17)	4	14% (5–20%)	11% (4–16%)	3	8% (5–9%)	13% (9–15%)	0	-
4. Dry finger foods and snacks (*n* = 14)	5	12% (8–13%)	9% (6–10%)	5	15% (13–18%)	24% (21–29%)	1	11%
5. Juices and other drinks (*n* = 2)	1	0% (--)	0% (--)	1	0% (--)	0% (--)	1	0% (--)
**Senegal**								
1. Dry, powdered, and instant cereal/starchy food (*n* = 76)	49	25% (18–55%)	20% (14–44%)	62	38% (32–47%)	60% (52–76%)	40	34% (32–50%)
2. Soft-wet spoonable, ready-to-eat foods (purees) (*n* = 194)	20	25% (16–25%)	20% (13–20%)	0	-	-	3	22% (20–22%)
3. Meals with chunky pieces (*n* = 49)	0	-	-	0	-	-	0	-
4. Dry finger foods and snacks (*n* = 10)	2	8% (8–8%)	6% (6–6%)	2	7% (7–7%)	11% (11–11%)	0	-
5. Juices and other drinks (*n* = 19)	0	-	-	0	-	-	0	-

^1^ Based on servings sizes declared on labels. 46 products without this label information were excluded from the analysis (Burkina Faso *n* = 2, Cameroon *n* = 4, Ghana *n* = 3, Nigeria *n* = 6, Senegal *n* = 31). ^2^ Three products excluded from the micronutrient analyses due to unclear iron/calcium/zinc content values reported on labels.

## Data Availability

Restrictions apply to the availability of these data. Data were obtained from Euromonitor International and Innova Market Insights and can be accessed from these third parties at https://www.euromonitor.com/ and https://www.innovamarketinsights.com/.

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
