# Peer review of "Nutrient Profiles of Commercially Produced Complementary Foods Available in Burkina Faso, Cameroon, Ghana, Nigeria and Senegal"

_nutrients, 2023, doi:10.3390/nu15102279_

Round 1

Reviewer 1 Report

the main question addressed by the research is To compare the composition of micronutrients in infant foods. The topic is relevant to the area of child nutrition.

The evaluation was carried out in regions that have not yet presented any studies on the subject. The methodology is suitable for the purposes of the study.  The conclusions are consistent with the evidence and arguments presented and authors address the main question posed.

references appropriate.

Author Response

Dear Nutrients,

We are pleased to resubmit for publication a revised manuscript of nutrients-2380682 entitled ‘Nutrient profiles of commercially produced complementary foods available in Burkina Faso, Cameroon, Ghana, Nigeria and Senegal countries’. Thank you very much for the consideration and comments from reviewers. We have responded to these comments as outlined below and relevant revisions are indicated with track changes in the resubmitted manuscript documents.

Best regards.

Reviewer 1

The main question addressed by the research is to compare the composition of micronutrients in infant foods. The topic is relevant to the area of child nutrition.

The evaluation was carried out in regions that have not yet presented any studies on the subject. The methodology is suitable for the purposes of the study.  The conclusions are consistent with the evidence and arguments presented and authors address the main question posed.

References appropriate.

Response:

Thank you for your review of our study on the composition of commercially produced complementary foods in West Africa. We appreciate your positive feedback regarding the relevance of the topic to child nutrition, as well as the suitability of our methodology for the study's purpose.

Reviewer 2 Report

This is an interesting study about Nutrient profiles of commercially produced complementary foods available in five areas. However, I have serious concerns about the interpretation of the section of introduction, methods and discussion in the present study.

Introduction

This study included four objectives.

However, you did not state clearly enough the necessity or reason for carrying out the objectives in the Introduction section.

The content in the current Introduction should be organized a little more compactly and the need for the present study should be added.

For example, the reasons for selecting these three nutrients for objective 1) are not stated. The need to consider the nutritional adequacy of CPCF in infant should be stated.

With regard to objective 3), is 'High sugar' an issue in the five regions? If so, you should explain this.

In order to investigate the objective 4), don't you need to determine the contents of the nutrients including objective 4) in the objective 1)?

Methods

L185-209: Why not create a Supplemental table of values used as standards?

Results

There is no coherence.

The results should not be enumerated, but rather organized in a straightforward manner so that they can be conveyed to the reader.

Discussion

L339-371: Are diseases such as those shown individually a problem in West Africa? How about adding a bit about that and connecting it to the need for measures to improve the situation?

If the use of the WHO Europe NPM as a reference was a limiting factor in this case, it would be better to discuss in the Discussion why this reference had to be used and at what points the use of this criterion requires caution in the interpretation of the results.

Have you asked a native English speaker to confirm this?

Author Response

Dear Nutrients,

We are pleased to resubmit for publication a revised manuscript of nutrients-2380682 entitled ‘Nutrient profiles of commercially produced complementary foods available in Burkina Faso, Cameroon, Ghana, Nigeria and Senegal countries’. Thank you very much for the consideration and comments from reviewers. We have responded to these comments as outlined below and relevant revisions are indicated with track changes in the resubmitted manuscript documents.

Best regards.

Reviewer 2

Introduction

This is an interesting study about Nutrient profiles of commercially produced complementary foods available in five areas. However, I have serious concerns about the interpretation of the section of introduction, methods and discussion in the present study.

For example, the reasons for selecting these three nutrients for objective 1) are not stated. The need to consider the nutritional adequacy of CPCF in infant should be stated.

Response: Many thanks for your valuable feedback on our manuscript. We have made several changes to the Introduction section to provide a clearer justification for the objectives of our study. We have also added a statement on the need to consider the nutritional adequacy of complementary foods for infants and young children, as well as the reasons for selecting the nutrients for objective 1 by elaborating on the prevalence of micronutrient deficiencies in West Africa as well as the growing issue of overnutrition.

With regard to objective 3), is 'High sugar' an issue in the five regions? If so, you should explain this.

Response:

Thank you for the comment. We agree that it is important to provide an explanation for why commercially produced complementary foods with high sugar content is of concern in the West African region. The high sugar in CPCF impacts the nutrient profile and adequacy of CPCFs. The issue of high sugar intake is closely related to the double burden of malnutrition, where undernutrition and overnutrition coexist. In the context of West Africa, the double burden of malnutrition is particularly concerning, as the region faces a high prevalence of undernutrition, including stunting, wasting, and micronutrient deficiencies, as well as a rising burden of overweight and obesity. High sugar intake is one of the factors contributing to the rising burden of overweight and obesity in the region, as it can lead to excessive calorie intake and weight gain, especially when combined with a sedentary lifestyle. At the same time, the consumption of high-sugar foods and drinks that are nutrient-poor can also displace nutrient-dense foods, leading to a lack of essential nutrients and contributing to undernutrition. Appropriate IYCF prevents undernutrition as well as obesity. Therefore, addressing the issue of high sugar intake, along with other dietary factors, is critical to addressing the double burden of malnutrition in West Africa and promoting optimal health outcomes for infants and young children in the region.

We have addressed this point in the Introduction section of the revised manuscript through the addition of further information on the challenges in West Africa, including the nutrition transition, double burden of malnutrition and increasing rates of overweight and noncommunicable diseases (NCDs). Additionally, the risk factors of NCDs, dental caries and overweight were previously stated in the Introduction.

In order to investigate the objective 4), don't you need to determine the contents of the nutrients including objective 4) in the objective 1)?

Response: The three micronutrients (iron, calcium, and zinc) have now been added to objective 1 as suggested.

Methods: L185-209: Why not create a Supplemental table of values used as standards?

Response: Thank you for your suggestion. We have now included this as a supplemental table as suggested.

Results. There is no coherence. The results should not be enumerated, but rather organized in a straightforward manner so that they can be conveyed to the reader

Response: We appreciate your suggestion regarding the organization of our results section. We have restructured the results section to improve coherence and readability, following a flow in line with the objectives and revising the placement of the tables and figures more logically within the text. We hope that these changes have addressed your concerns.

Discussion: L339-371: Are diseases such as those shown individually a problem in West Africa? How about adding a bit about that and connecting it to the need for measures to improve the situation?

Response:

Thank you for your comment. We have included further details on this in the Discussion section and on how the promotion of appropriate infant feeding practices can help prevent these diseases. We believe that this addition strengthens the article by providing a more comprehensive picture of the health challenges facing West Africa and the potential impact of infant feeding interventions in addressing these challenges.

If the use of the WHO Europe NPM as a reference was a limiting factor in this case, it would be better to discuss in the Discussion why this reference had to be used and at what points the use of this criterion requires caution in the interpretation of the results.

 Response:

Thank you for the comment on the use of the WHO Europe NPM as a reference in our study. We have taken your comments into consideration and revised the manuscripts within the Discussion section. As previously noted in the Discussion, this is the only NPM for CPCF that exists globally and so this model was used. We believe that this discussion will provide a better understanding of the limitations of our study and the implications of using this the WHO EU NPM as reference.

Comments on the Quality of English Language. Have you asked a native English speaker to confirm this?

Response:

Thank you for your comments. We have reviewed the manuscript and made necessary corrections to improve the language quality, as suggested. Our corresponding author is also a native English speaker who contributed to the write up and has provided a comprehensive review of this manuscript, both in its original and revised state.

Round 2

Reviewer 2 Report

Correction confirmed.